# Genotypic glucose-6-phosphate dehydrogenase (G6PD) deficiency protects against *Plasmodium falciparum* infection in individuals living in Ghana

**Linda Eva Amoah**[1]*, **Kwame Kumi Asare**[2], **Donu Dickson**[1], **Joana Abankwa**[1], **Abena Busayo**[1], **Dorcas Bredu**[1], **Sherifa Annan**[1], **George Adu Asumah**[3], **Nana Yaw Peprah**[3], **Alexander Asamoah**[3], **Keziah Laurencia Malm**[3]

**1** Dept. of Immunology, Noguchi Memorial Institute of Medical Research, University of Ghana, Accra, Ghana, **2** Dept. of Biomedical Science, School of Allied Health Sciences, College of Allied Health Sciences, University of Cape Coast, Cape Coast, Ghana, **3** National Malaria Control Program, Accra, Ghana

* lek.amoah@gmail.com, lamoah@noguchi.ug.edu.gh

**Data Availability Statement:** All relevant data are within the manuscript and its Supporting Information files.

## Abstract

### Introduction

The global effort to eradicate malaria requires a drastic measure to terminate relapse from hypnozoites as well as transmission via gametocytes in malaria-endemic areas. Primaquine has been recommended for the treatment of *P. falciparum* gametocytes and *P. vivax* hypnozoites, however, its implementation is challenged by the high prevalence of G6PD deficient (G6PDd) genotypes in malaria endemic countries. The objective of this study was to profile G6PDd genotypic variants and correlate them with malaria prevalence in Ghana.

### Methods

A cross-sectional survey of G6PDd genotypic variants was conducted amongst suspected malaria patients attending health care facilities across the entire country. Malaria was diagnosed using microscopy whilst G6PD deficiency was determined using restriction fragment length polymorphisms at position 376 and 202 of the G6PD gene. The results were analysed using GraphPad prism.

### Results

A total of 6108 subjects were enrolled in the study with females representing 65.59% of the population. The overall prevalence of malaria was 36.31%, with malaria prevalence among G6PDd genotypic variants were 0.07% for A-A- homozygous deficient females, 1.31% and 3.03% for AA- and BA- heterozygous deficient females respectively and 2.03% for A- hemizygous deficient males. The odd ratio (OR) for detecting *P. falciparum* malaria infection in the A-A- genotypic variant was 0.0784 (95% CI: 0.0265–0.2319, p<0.0001). Also, *P. malariae* and *P. ovale* parasites frequently were observed in G6PD B variants relative to G6PD A-variants.

**Funding:** KM, Grant from the Global Fund through the Ghana National Malaria Control Program. The funders had no role in the study design, data collection and analysis, decision to publish, or preparation of the manuscript.

**Competing interests:** The authors have declared that no competing interests exist.

## Conclusion

G6PDd genotypic variants, A-A-, AA- and A- protect against *P. falciparum*, *P. ovale* and *P. malariae* infection in Ghana.

## Introduction

*Plasmodium falciparum* malaria is the most lethal disease estimated to cause 400,000 deaths annually in sub-Saharan Africa [1, 2]. *Humans* have been under evolutionary and or selective pressure due to malaria infection [3]. The geographical distribution of erythrocyte genetic polymorphisms in malaria endemic countries have been suggested to provide survival advantages against severe and complicated forms of malaria [3, 4]. The red blood cells which support the erythrocytic life cycle of *Plasmodium* parasites have evolved with diverse genetic polymorphisms such as various hemoglobinopathies, including the sickle cell trait and α/β-thalassemia's as well as the selection of various blood group antigen variants such as blood group O that confer protective advantages against cerebral malaria and other disease states of malaria [5, 6]. Glucose-6-phosphate dehydrogenase deficiency (G6PDd) is another erythrocyte polymorphism reported to offer protection against malaria. G6PDd has a similar geographical distribution as *P. falciparum* and *P. vivax* infections [7–9].

Globally, about 400 million people are affected with more than 400 variants of G6PDd caused by 200 known mutations in the gene encoding the human G6PD [10]. The enzymopathy, G6PDd is an X-linked disease associated with various degrees of reduced G6PD enzyme activities [11]. A single amino acid substitution at position A376G converts the genotypic variant B to variant A, and a second amino acid substitution at position G202A result in a change from the A variant into the A- (A376G/G202A) variant [12, 13]. The B, A, A- G6PDd variants are the most predominant in sub-Saharan Africa [14–16]. The B variant exhibits complete G6PD enzyme activity, the A variant possesses about 85% G6PD enzymatic activities, whiles the A- variants exhibit 10% enzymatic activity [17, 18]. Other A- variants such as A376G/A543T, A376G/G680T and A376G/T968C are also present in some countries in sub-Saharan Africa [17]. The heterozygotes A-A- trait has been suggested to provide a selective advantage against severe malaria in females [19–21]. The protection offered by G6PDd A- against malaria infection is not well understood. It is unclear however, if both hemizygotes (A-) and heterozygotes (AA-) offer protection against severe malaria or bias towards females [9, 22]. Although a recent case-control study conducted in Africa has reported a 46–58% reduced risk in severe malaria by both hemizygotes and heterozygotes G6PDd A- [23] and heterozygotes G6PDd A- but not hemizygotes G6PDd A- has been reported to protect against acute malaria among children in Nigeria, suggesting sex-specific malaria protection [24, 25]. The variations in the inactivation of enzymatic activities observed among the G6PDd phenotypes is a challenge for assessing the protective effects offered by G6PDd against malaria [26, 27].

Host-parasite interactions result in a complex balance of pro-oxidant and antioxidant molecules in both host and parasite [28]. However, the G6PDd erythrocytes may not sufficiently counterbalance oxidative stress generated by the erythrocytes stage malaria parasites observed in drugs such as primaquine, dapsone, sulfonamides, quinolones, chloramphenicol, nitrofurantoin (antibiotics), and phenazopyridine (analgesics) used by affected individuals [29]. G6PDd decreases the risk of cerebral malaria but increases the risk of severe anaemia in malaria infection [10, 30].

The global effort on eradication of malaria requires drastic measures to terminate relapse from hypnozoites and kill off gametocytes in the malaria-endemic areas [31, 32]. However, primaquine (the only 8-aminoquinoline) recommended for the treatment and elimination *of P. vivax* hypnozoites and *P. falciparum* gametocytes has limited usage among G6PDd conditions [33]. The implementation of the recommendation to use primaquine in malaria-endemic regions with the high prevalence of G6PDd phenotypes is a challenge [32, 34]. As primaquine can trigger oxidative stress in G6PDd affected erythrocytes, making them more susceptible to oxidative haemolysis and subsequently increases physiopathogenesis in the host [8].

The geographical distribution of G6PDd in the malaria-endemic regions suggests that it is naturally selected to offer protection against severe forms of malaria [16]. The global prevalence of G6PDd based on DNA analysis is estimated at 7.1%, with Africa estimated to have a prevalence of 24%, [35] and a prevalence of 12.4% in Ghana [17]. However, the protective role of G6PDd remains unresolved as hemizygotes G6PDd A- males seem unprotected from severe malaria although majority of heterozygote deficient A- females are protected [8, 21]. The objective of this study was to identify the prevalence and distribution of G6PD genotypic variants in Ghana as well as determine whether G6PDd genotypic variants protect against symptomatic malaria infection.

## Methods

### Study site and sampling

Samples used for this study were collected in 2018, from 10 health facilities scattered across each of the 10 regions of Ghana, making a total of 100 facilities (S1 File) as part of a cross-sectional study that recruited 19896 suspected malaria patients [36, 37]. The study was performed in accordance with the Declaration of Helsinki. Ethical approval (# 068/17-18) was obtained from the Institutional Review Board, NMIMR, University of Ghana. Written parental consent was obtained from parents or guardians for all the children recruited in this study. All children aged 12 years old and above were also made to endorse a child assent form. All methods were carried out in accordance with relevant guidelines and regulations.

A total volume of 2 ml of whole blood was collected and an aliquot was used to prepare dried filter paper blood spots (DBS) and prepare thick and thin blood films. Thick and thin blood films were prepared from the original blood samples according to standard protocols [38–40] and read by two WHO certified expert microscopists, with all discordant results clarified by a third WHO certified malaria microscopist. Parasite densities from each slide were determined by multiplying the total number of parasites identified per 200 white blood cells by 40. Stratified random sampling method was applied to the recruited samples where each study region was considered as an individual group and randomly selected from each region to ensure equal representation in the sampling for the study. A total sample of 6108 participants were randomly selected from all the regions and for use in this study. The demographic details (age and sex) and the infecting malaria parasite species of the participants whose samples were used in this study were obtained from the larger study [36, 37].

### DNA extraction from dried blood blots

DNA was extracted from two 3 mm disks of filter paper dried blood spots (DBS) using the saponin chelex method [41, 42]. Briefly, 1 ml of 1X Phosphate Buffered Saline (PBS) supplemented with 0.5% Tween was added to the microcentrifuge tube containing the disks. The tube was vortexed briefly and incubated overnight at room temperature with shaking. The tube was then centrifuged for 2 mins at 14,000 rpm and the supernatant decanted. Subsequently, 1 ml of ice-cold 1X PBS was added to the tube and incubated at 4°C for 30 mins. The

tube was spun at 14,000 rpm for 2 min followed by aspiration of the supernatant. Finally, 150 μl of 13.3% chelex-100 in water was then added to the tube and incubated at 95°C for 10 mins. The tube was then spun for 8 mins at 14,000 rpm and the supernatant containing the DNA was aliquoted into a sterile labeled 0.5 ml microfuge tube and either used immediately or stored at -20°C.

## G6PD genotyping

A protocol similar to that reported by Amoah *et al.* [17] was used for this. Briefly, the A376G polymorphism was genotyped by PCR followed by restriction fragment length polymorphism. In the PCR reaction, 3 μl of each of the extracted DNA was amplified in a 10 μL reaction consisting of 1X AmpliTaq Gold® Fast PCR Master Mix, UP (Applied Biosystems, USA), 0.2 μM of forward and reverse primers A376GF and A376GR respectively. The cycling conditions for the PCR included a polymerase activation step at 95°C for 10 minutes followed by 35 cycles of denaturation step at 96°C for 30 seconds, primer annealing step at 61°C and extension at 68°C for 30 seconds. This was followed by a final extension at 72°C for 10 minutes. The amplicons from the PCR reaction were digested in a 15 μl reaction containing 8 μl of amplicon and 1 unit of Fok1 (New England Biolabs, UK). The digest reaction was incubated at 37°C for 1 hour and products resolved on a 2% (w/v) agarose gel pre-stained with Ethidium bromide. The agarose gel was visualized under UV using a Vilber gel documentation system.

All samples that yielded a digested product for the A376G reaction were further selected for the genotyping of the G202A polymorphism, using similar PCR conditions as the A376G but with the primers G202AF and G202AR (forward and reverse primers respectively) and an annealing temperature of 65°C. The PCR amplicons were digested with 1 unit of NlaIII (New England Biolabs, UK) under the same conditions as Fok1 but with an incubation period of 20 minutes. The digest products were resolved just as indicated above.

Lambda DNA was used as a positive control for both the FokI and the NlaIII digestion reactions. Distilled water was used as a negative control template for the A376G and G202A PCR reactions.

## Statistical analysis

All the data acquired from the study were entered into Microsoft Excel (Microsoft software) and the statistical analyses were performed with GraphPad Prism software, version 8.4.3 (GraphPad Software). The data were grouped based on G6PD genotypic variants and variants prevalence was determined. The prevalence of malaria parasites among the individual G6PD genotypic variants was determined. Association of malaria infection and G6PD genotypic variants were tested using odds ratio, receiver-operation characteristics (ROC) curve, type III ANOVA, Turkey's multiple comparison test and restriction cubic spline curve statistical tests were used to analyze data. Statistically significant were considered at p-value < .05.

## Results

### Demographic characteristics of the study subjects

A total of 6108 subjects from 10 regions of Ghana were enrolled in the study. The Eastern region had the highest number of subjects (N = 787) enrolled in the study, out of which 34.31% were male and 65.59% were female. The Volta region had the least number (N = 465) of subjects out of which 41.51% were male and 58.49% female (Table 1).

**Table 1. Demographic characteristics of study subjects.**

| | Brong Ahafo | Ashanti | Eastern | Central | Northern | Greater Accra | Upper West | Upper East | Volta | Western |
|---|---|---|---|---|---|---|---|---|---|---|
| **Sex, n (%)** | | | | | | | | | | |
| Male | 305 (41.50) | 299 (40.57) | 270 (34.31) | 184 (37.25) | 213 (38.31) | 245 (41.88) | 240 (35.61) | 190 (31.83) | 193 (41.51) | 213 (44.56) |
| Female | 430 (58.50) | 438 (59.43) | 517 (65.69) | 310 (62.75) | 343 (61.69) | 340 (58.12) | 434 (64.39) | 407 (68.17) | 272 (58.49) | 265 (55.44) |
| **Age mean (SEM), yrs** | | | | | | | | | | |
| Male | 12.93 (0.98) | 13.99 (0.91) | 18.50 (1.13) | 13.44 (1.09) | 14.38 (1.14) | 20.69 (1.07) | 18.52 (1.26) | 22.11 (1.54) | 14.69 (0.77) | 18.64 (1.28) |
| Female | 21.61 (1.02) | 20.82 (0.87) | 29.43 (0.95) | 23.43 (1.07) | 21.42 (0.96) | 28.01 (0.99) | 24.85 (0.96) | 30.34 (1.14) | 19.87 (0.77) | 21.72 (0.95) |
| **Malaria + n (%)** | | | | | | | | | | |
| Male | 152 (49.84) | 128 (42.81) | 15 (5.56) | 105 (57.07) | 67 (31.46) | 55 (22.45) | 93 (38.75) | 67 (35.26) | 127 (65.80) | 135 (63.38) |
| Female | 179 (41.63) | 186 (42.47) | 36 (6.96) | 186 (60.00) | 99 (28.86) | 71 (20.88) | 129 (29.72) | 86 (21.13) | 154 (56.62) | 151 (56.98) |
| Total | 331 (45.03) | 314 (42.61) | 51 (6.48) | 291 (58.91) | 166 (29.86) | 126 (21.54) | 219 (32.49) | 153 (25.63) | 281 (60.43) | 286 (59.83) |

### Glucose-6-phosphate dehydrogenase deficiency (G6PDd) genotypic variant A-A- protects against malaria infections in malaria-endemic regions in Ghana

The total prevalence of malaria across the 10 regions of Ghana was 2,218 (36.31%) with 60.43% recorded at Volta and 6.48% recorded in the Eastern region (Table 1). The prevalence of malaria parasites in samples with G6PDd A-A- (0.07%), AA- (1.31%) and A- (2.03%) was low (Fig 1A and Table 2). Similarly, the overall prevalence of malaria in A- and BA- genotypic variants was 2.26% and 3.03% respectively (Table 2). The prevalence of the G6PD A- genotype ranged from 13.97% in the Central region to 25.37% in the Upper West region. The heterozygous G6PD A- (AA-, BA-) was highest in the Greater Accra region (17.78%) and lowest in the Central region (9.31%). Also, the homozygous G6PD A- (A-A-) ranged from 4.66% in the Central region to 7.74% in the Western region (Table 3).

The odds of detecting malaria via microscopy among the A- variant was 0.764 (0.5818–1.0035), p = 0.0453 and 0.4841 (0.3917–0.5988), p<0.0001 in the BA- variant (Table 4). The accuracy of detecting malaria parasites among the individual G6PDd genotypic variants was tested using the receiver operating characteristic (ROC) curve (Fig 1B). This showed that there were high specificity and sensitivity in the association between malaria infection and the G6PDd genotypic variants. The likelihood ratio for the association between malaria and G6PDd genotypic variant ranged from 1.33 to 5.00 with sensitivity (95% CI) of 72.73 (43.44–90.25) to 45.45 (21.27–71.99) and specificity (95% CI) of 45.55 (21.27–71.99) to 90.91 (62.26–99.53). The highest sensitivity of 72.73 (43.44–90.25) and specificity of 72.73 (43.44–90.25) with a likelihood ratio of 2.667 (Fig 1B and S1 Table). The homozygote G6PDd A-A- female exhibited a stronger protection against malaria OR (95% CI) = 0.00784 (0.0265–0.2319), p<0.0001 compared to the heterozygote G6PDd AA- female OR (95% CI) = 0.6741 (0.4878–0.9316), p = 0.0169 and the hemizygote G6PDd A- male OR (95% CI) = 0.764 (0.5818–1.0035), p = 0.0453 (Table 4). The type III ANOVA statistics showed a significant difference among the G6PDd genotypic variants and their association of malaria infection cases and malaria negative cases samples among the study subjects (S2 Table).

The association of G6PDd genotypic variants, malaria infections and age were modelled using restriction cubic spline curve for the stratified age categories of ≥ 20 years, 15–19 years, 10–14 years, 5–9 years and 0–4 years. The result showed a high malaria infection among B types G6PDd genotypic variants with increasing expansion of A-, AA- and A-A- genotypic variants which offered protection against malaria infections at lower age categories of 0–4 years and 5–9 years compared to the other age categories (S1A and S1B Fig). Highly significant

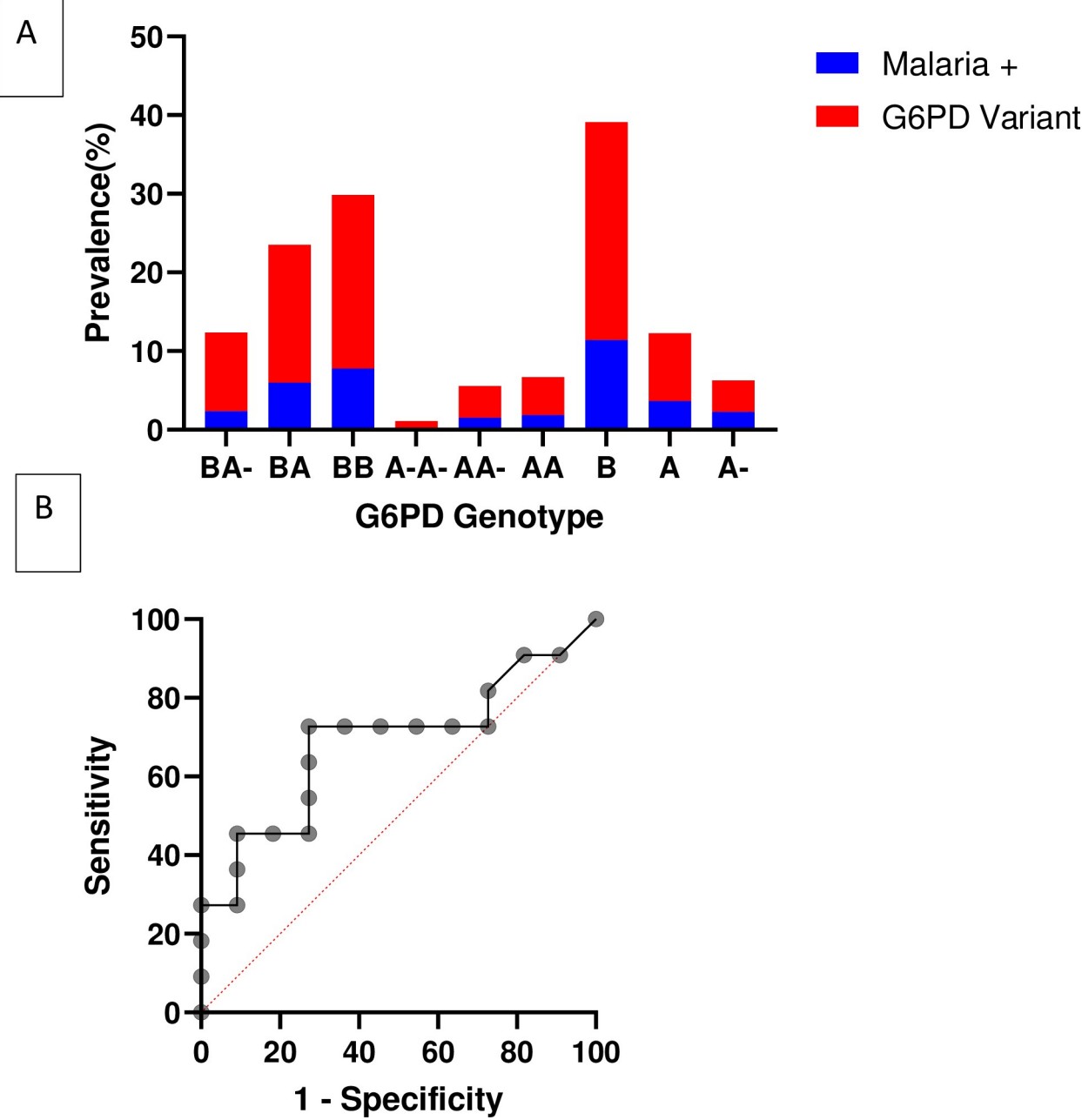

**Fig 1. Malaria prevalence among G6PD genotypic variants.** A) Prevalence of individual G6PD genotypic variants and their corresponding malaria prevalence. B) Specificity and Sensitivity of detecting malaria infection among the different G6PD genotypic variants using ROC.

protection against malaria infection was observed in A-A- genotypic variant (S2 File). The associations observed between the G6PDd genotypic variants, malaria infections and age were further analysed using scatterplots of residuals at the different restricted cubic spline knots at 1, 2, 3, 4 & 5, random intercepts and slopes to assess the goodness-of-fit to capture the heterogeneity in the selection of G6PDd genotypic variants in malaria infection-age curves among the associations. The display of residuals implied that variance of the error was standardized across age (S1C Fig). The sigmoidal 4PL of least squares fit analysis showed that the R squared

**Table 2. The regional distribution of G6PD genotypic variants and *Plasmodium falciparum* in Ghana.**

| Genotypic variants | Brong Ahafo | | | Ashanti | | | Eastern | | | Central | | | Northern | | | Greater Accra | | | Upper West | | | Upper East | | | Volta | | | Western | | |
|---|---|---|---|---|---|---|---|---|---|---|---|---|---|---|---|---|---|---|---|---|---|---|---|---|---|---|---|---|---|---|
| | G6PD | P. falciparum n (%) | other malaria | G6PD | P. falciparum n (%) | other malaria | G6PD | P. falciparum n (%) | other malaria | G6PD | P. falciparum n (%) | other malaria | G6PD | P. falciparum n (%) | other malaria | G6PD | P. falciparum n (%) | other malaria | G6PD | P. falciparum n (%) | other malaria | G6PD | P. falciparum n (%) | other malaria | G6PD | P. falciparum n (%) | other malaria | G6PD | P. falciparum n (%) | other malaria |
| **Female** | | | | | | | | | | | | | | | | | | | | | | | | | | | | | | |
| BB | 18 (2.45) | 7 (2.18) | - | 109 (14.79) | 33 (10.61) | - | 48 (6.10) | 11 (21.57) | - | 80 (16.19) | 63 (22.50) | - | 41 (7.37) | 39 (28.26) | - | 12 (2.05) | 7 (5.74) | - | 19 (2.82) | 5 (2.46) | - | 9 (1.51) | 7 (4.83) | - | 25 (5.38) | 19 (8.15) | 3 (15.79) | 7 (1.46) | 7 (2.49) | - |
| BA- | 62 (8.44) | 24 (7.48) | - | 63 (8.55) | 33 (10.61) | - | 68 (8.64) | 8 (15.69) | - | 37 (7.49) | 14 (5.00) | 2 (66.67) | 69 (12.41) | 15 (10.87) | 2 (66.67) | 81 (13.85) | 20 (16.39) | - | 85 (12.61) | 28 (13.79) | - | 63 (10.55) | 14 (9.66) | - | 37 (7.96) | 14 (6.01) | 3 (15.79) | 40 (8.37) | 17 (6.05) | 1 (33.33) |
| BA | 119 (16.19) | 62 (19.31) | - | 110 (14.93) | 56 (18.01) | - | 169 (21.47) | 9 (17.64) | - | 87 (17.61) | 69 (24.64) | - | 115 (20.68) | 2 (1.45) | 3 (75.00) | 90 (15.38) | 17 (13.93) | - | 126 (18.69) | 28 (13.79) | 2 (28.57) | 120 (20.10) | 22 (15.17) | - | 67 (14.41) | 32 (13.73) | 3 (15.79) | 63 (13.18) | 38 (13.52) | - |
| A-A- | 5 (0.68) | - | - | 4 (0.54) | - | - | 8 (1.02) | 1 (1.96) | - | 2 (0.40) | - | - | 5 (0.90) | 1 (0.72) | - | 3 (0.51) | - | - | 35 (5.19) | 2 (0.99) | - | 5 (0.84) | - | - | 6 (1.29) | - | - | 2 (0.42) | - | - |
| AA- | 26 (3.54) | 13 (4.05) | - | 44 (5.97) | 23 (7.40) | - | 23 (2.92) | 2 (3.92) | - | 9 (1.82) | 3 (1.07) | - | 26 (4.68) | 5 (3.63) | - | 23 (3.93) | 9 (7.38) | - | 6 (0.89) | 6 (2.96) | - | 18 (3.02) | 5 (3.45) | - | 6 (1.29) | 6 (2.58) | - | 26 (5.44) | 14 (4.98) | - |
| AA | 37 (5.03) | 18 (5.61) | - | 41 (5.56) | 23 (7.40) | - | 49 (6.23) | 3 (5.88) | - | 17 (3.44) | 13 (4.64) | - | 11 (1.98) | 6 (4.35) | - | 29 (4.96) | 5 (4.10) | - | 34 (5.04) | 9 (4.43) | - | 26 (4.36) | 10 (6.90) | - | 15 (3.23) | 8 (3.43) | 1 (5.26) | 28 (5.86) | 20 (7.12) | - |
| **Male** | | | | | | | | | | | | | | | | | | | | | | | | | | | | | | |
| B | 353 (48.03) | 129 (40.19) | 1 (100.0) | 256 (34.73) | 76 (24.43) | 1 (33.33) | 321 (40.79) | 11 (21.57) | - | 206 (41.70) | 79 (28.21) | 1 (33.33) | 208 (37.41) | 50 (36.23) | - | 242 (41.37) | 42 (34.43) | - | 263 (39.02) | 90 (44.33) | 4 (57.14) | 286 (47.91) | 66 (45.52) | 1 (100) | 263 (56.56) | 153 (65.67) | 7 (36.84) | 243 (50.84) | 142 (50.53) | 2 (66.67) |
| A- | 49 (6.67) | 29 (9.03) | - | 36 (4.88) | 23 (7.40) | - | 39 (4.96) | 1 (1.96) | - | 21 (4.25) | 10 (3.57) | - | 35 (6.29) | 10 (7.25) | 1 (25.00) | 38 (6.50) | 8 (6.50) | - | 45 (6.68) | 18 (8.87) | 1 (14.29) | 34 (5.70) | 9 (6.21) | - | 16 (3.44) | - | - | 25 (5.23) | 16 (5.69) | - |
| A | 66 (8.98) | 39 (12.15) | - | 74 (10.04) | 44 (14.15) | - | 62 (7.89) | 5 (9.80) | - | 35 (7.09) | 29 (10.36) | - | 46 (8.27) | 10 (7.25) | - | 67 (11.45) | 14 (11.48) | - | 61 (9.05) | 17 (8.37) | - | 36 (6.03) | 12 (8.28) | - | 30 (6.45) | 1 (0.43) | 2 (10.53) | 44 (9.21) | 27 (9.61) | - |
| Total | 735 (12.07) | 321 (15.78) | 1 (2.44) | 737 (12.07) | 311 (15.293) | 3 (7.32) | 787 (12.88) | 51 (2.51) | - | 494 (8.09) | 280 (13.77) | 3 (7.32) | 556 (9.58) | 138 (6.78) | 4 (9.76) | 585 (9.58) | 122 (6.00) | - | 674 (11.03) | 203 (9.98) | 7 (17.07) | 597 (9.77) | 145 (7.13) | 1 (2.44) | 465 (7.61) | 233 (11.46) | 19 (46.34) | 478 (7.83) | 281 (13.81) | 3 (7.32) |

**Table 3. The regional distribution of individuals with varying G6PD genotypic variants in Ghana.**

|  | Ashanti, n (%) | Brong Ahafo, n (%) | Eastern, n (%) | Central, n (%) | Greater Accra, n (%) | Northern, n (%) | Upper West, n (%) | Upper East, n (%) | Volta, n (%) | Western, n (%) |
|---|---|---|---|---|---|---|---|---|---|---|
| Female |  |  |  |  |  |  |  |  |  |  |
| BB | 109 (14.79) | 18 (2.45) | 48 (6.1) | 80 (16.19) | 12 (2.05) | 41 (7.37) | 19 (2.82) | 9 (1.51) | 25 (5.38) | 7 (1.46) |
| BA- | 63 (8.55) | 62 (8.44) | 68 (8.64) | 37 (7.49) | 81 (13.85) | 69 (12.41) | 85 (12.61) | 63 (10.55) | 37 (7.96) | 40 (8.37) |
| BA | 110 (14.93) | 119 (16.19) | 169 (21.47) | 87 (17.61) | 90 (15.38) | 115 (20.68) | 126 (18.69) | 120 (20.10) | 67 (14.41) | 63 (13.18) |
| A-A- | 4 (0.54) | 5 (0.68) | 8 (1.02) | 2 (0.4) | 3 (0.51) | 5 (-0.9) | 35 (5.19) | 5 (0.84) | 6 (1.29) | 2 (0.42) |
| AA- | 44 (5.97) | 26 (3.54) | 23 (2.92) | 9 (1.82) | 23 (3.93) | 26 (4.68) | 6 (0.89) | 18 (3.02) | 6 (1.29) | 26 (5.44) |
| AA | 41 (5.56) | 37 (5.03) | 49 (6.23) | 17 (3.44) | 29 (4.96) | 11 (1.98) | 34 (5.04) | 26 (4.36) | 15 (3.23) | 28 (5.86) |
| Male |  |  |  |  |  |  |  |  |  |  |
| B | 256 (34.73) | 353 (48.03) | 321 (40.79) | 206 (41.70) | 242 (41.37) | 208 (37.41) | 263 (39.02) | 286 (47.91) | 263 (56.56) | 243 (50.84) |
| A- | 36 (4.88) | 49 (6.67) | 39 (4.96) | 21 (4.25) | 38 (6.50) | 35 (6.29) | 45 (6.68) | 34 (5.70) | 16 (3.44) | 25 (5.23) |
| A | 74 (10.04) | 66 (8.98) | 62 (7.89) | 35 (7.09) | 67 (11.45) | 46 (8.27) | 61 (9.05) | 36 (6.03) | 30 (6.45) | 44 (9.21) |

value of genotypic variants of A (0.7039), A- (0.7143) and AA- (0.5806) and a span (95% CI) of 20.93 (-2.08e+50 - ∞), 6.00 (-76.88 - ∞) and 96.50 (-782.9 - ∞) for A, A- and AA- respectively across the age categories shows a low malaria infection rate compared to the B or BB, BA, BA- genotypic variants (S1C Fig).

## *Plasmodium malariae* and *Plasmodium ovale* preferentially invade G6PDd B variant

Females with the BB, BA & BA- G6PD genotypic variants had a higher prevalence of Plasmodium falciparum infection relative to the females with the AA, AA- A-A- variants, whilst males with the B genotype had higher infection prevalence relative to those with the A, & A- variants.

**Table 4. The odds of detecting malaria microscopically among the glucose-6-phosphate dehydrogenase genotypic variants.**

| G6PD | Malaria N (%) | OR (95% CI) | z-statistic | p |
|---|---|---|---|---|
| Males |  |  |  |  |
| A | 216 (3.58) | 0.8151 (0.6543–1.0153) | 1.824 | 0.0681 |
| A- | 136 (2.26) | **0.764 (0.5818–1.0035)** | **1.935** | **0.0453** |
| B | 675 (11.19) | 0.8045 (0.7108–0.9106) | 3.441 | 0.0006 |
| Females |  |  |  |  |
| AA | 111 (1.84) | 0.74 (0.5485–0.9983) | 1.971 | 0.0487 |
| AA- | 91 (1.51) | 0.6741 (0.4878–0.9316) | 2.389 | 0.0169 |
| **A-A-** | **4 (0.07)** | **0.0784 (0.0265–0.2319)** | **4.602** | **<0.0001** |
| BB | 462 (7.66) | 0.5992 (0.5211–0.6891) | 7.183 | <0.0001 |
| BA | 353 (5.85) | 0.5603 (0.4787–0.6559) | 7.21 | <0.0001 |
| BA- | 183 (3.03) | 0.4841 (0.3917–0.5983) | 6.714 | <0.0001 |

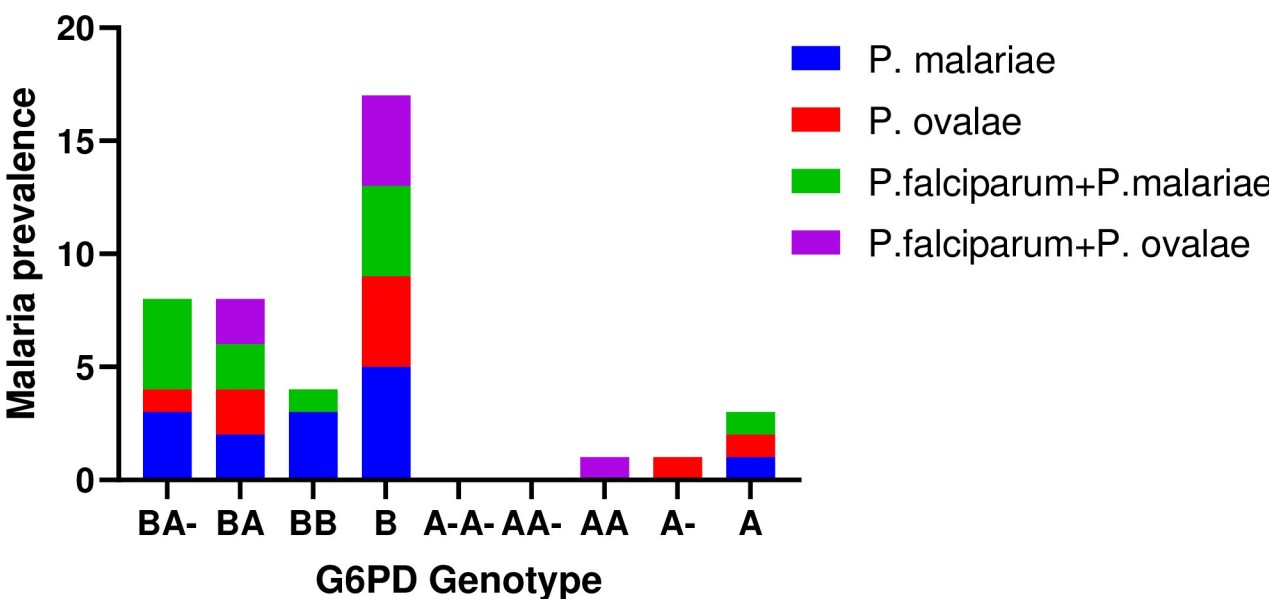

**Fig 2. The prevalence of non-falciparum and or mixed falciparum infection among G6PD genotypic variants in Ghana.**

Males with the B genotype had the highest prevalence of P. falciparum infection (N = 675) followed by females with the BB variant (N = 418). To control the false discovery rate between G6PDd genotypic variants and P. falciparum malaria was tested using a two-stage-up procedure of Benjamini, Krieger & Yekutieli test without assuming a constant standard deviation. The results showed a significant difference in the means of P. falciparum infections between the individual G6PDd genotypic variants with A vs B (-287.40; p<0.0398), A vs A- (44.73; p<0.0284), & A- vs B (-332.20; p<0.0185) among the male subjects and A-A- vs BB (-633.30; p<0.0166) & AA- vs BB (-543.40; p<0.001) among the female subjects. Again, Plasmodium malariae and Plasmodium ovale malaria which is less prevalent in Ghana preferentially infected BB, BA, BA- in females and the B genotype in males, with only a few infections identified in males with A, A- genotypes and females with the AA genotype (Fig 2).

### The pattern of distribution of G6PDd genotypic variants and malaria distribution across the ten regions in Ghana

The prevalence of the different G6PDd variants was significantly different ($\chi^2$ = 60.77, p<0.0001) but this was very similar in the 10 regions ($\chi^2$ = 6.005, p = 0.7344 for A-A-; $\chi^2$ = 5.243, p = 0.8127 for AA-; $\chi^2$ = 1.677, p = 0.9956 for A- and $\chi^2$ = 3.694, p = 0.9304 for BA-). Plasmodium falciparum infection follows a similar pattern of the distribution of G6PDd genotypic variants except for P. falciparum infection patterns observed in the Eastern region of Ghana (Fig 3A). The A-A- genotypic variant consistently showed protection against malaria infections across the 10 regions of Ghana (Fig 3B). The overall prevalence of P. malariae (1.22%) and P. ovale (0.72%) were recorded in the study. The P. malariae parasites were uniformly distributed in 8 out of the 10 regions, except for the Eastern and Greater Accra region. Whereas P. ovale parasites were observed in four regions, Brong Ahafo, Upper West, Volta and Western regions of Ghana (Fig 3C).

### Discussion

The distribution and selection of genetic variants of the G6PD gene in malaria-endemic regions and its associated protection against severe malaria remain controversial [41–43]. No

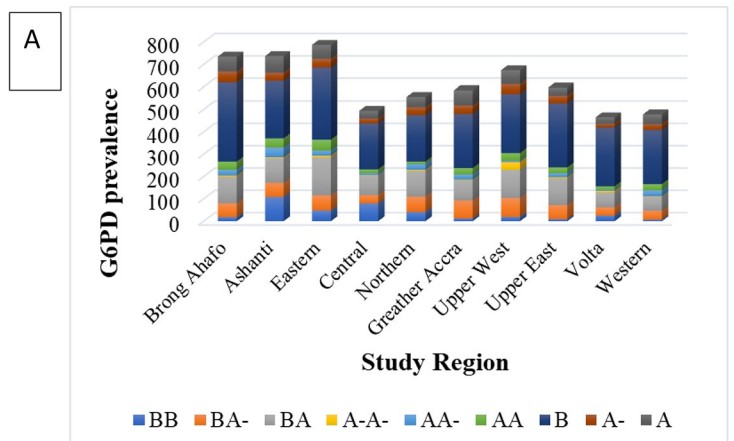

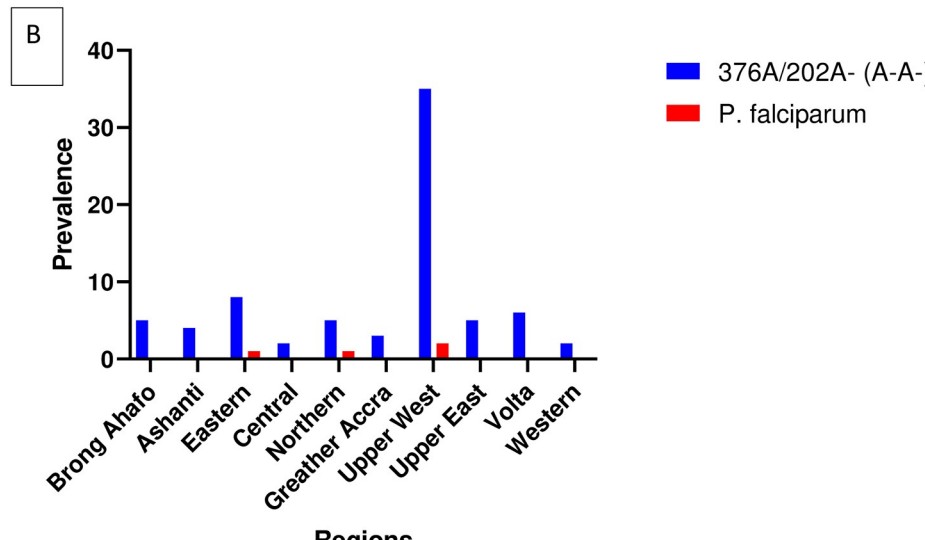

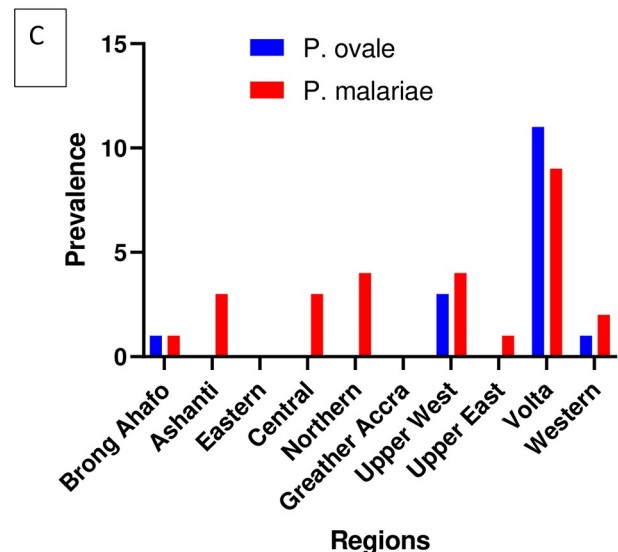

**Fig 3. The regional distribution of G6PD genotypic variants and their associated *P. falciparum* infections in Ghana. A.** The pattern of distribution of the G6PD prevalence across the 10 regions of Ghana. **B.** The G6PD genotypic variant A-A- protect against *P. falciparum* malaria infection across the 10 regions of Ghana. **C.** The regional distribution of *P. ovale* and *P. malariae* in Ghana.

specific G6PDd allelic genotype has been selected for balancing selection and fixed advantage against severe malaria [10, 30].

A nationwide prevalence of G6PDd genotypic variants based on the polymorphisms at 376 and 202 positions of the coding region of the human G6PD gene and the pattern of malaria infections was conducted. The result showed that G6PDd genotypic variants were homogenous throughout Ghana. The prevalence of G6PDd A- variants (including male hemizygous, female homozygous and heterozygous) recorded in this study (1225/6108, 20.06%) is similar to the previous reports from Ghana and other parts of West Africa [15, 44–47].

This study also assessed the association of G6PDd A- variants and protection against malaria in Ghana. The individual G6PDd A- variants (BA- and AA- heterozygous female, A-A- homozygous female and A- hemizygous male) and the pattern of malaria infections analyzed showed a significant association. The results showed that both hemizygous male (A-) and heterozygous female (AA-/BA-) G6PDd variants had significantly lower frequencies of malaria infections. This result agrees with previous reports that A- male and AA-/BA- female G6PDd variants protect P. falciparum-infected individuals from severe malaria [16, 48]. Although there were significant associations between both the BA- and AA- heterozygous alleles and the protection against malaria [49], the AA- heterozygous allele offered stronger protection against malaria than the BA- allele. Previous reports that treated and analyzed the two females heterozygous G6PDd genotypes (BA- and AA-) as a combined genotype did not find any association between the female heterozygous G6PDd A- and protection against malaria [19, 20, 48].

Also, the female homozygous A-A- G6PDd variant was highly associated with protection against malaria infections. The protective effects of A-A- homozygous was observed throughout all the regions in Ghana. The prevalence of the G6PDd variant A-A- was 0.42% in the Western and 5.19% in the Upper West region in Ghana. The A-A- homozygous deficiency at a prevalence of 0.5% offer protection against severe malaria among female in the Gambia [50, 51]. The rarely reported association of homozygous A-A- and protection against malaria has to do with the limited prevalence for statistical power for correlation analysis. The observation made in this study is an indication that A-A- female homozygous G6PDd A- variants may offer higher protection against malaria infection compared to hemizygous and heterozygous G6PDd A- variants. The reason for the low rise in frequency of the A-A- homozygous allele for the positive selection of malaria remains obscured. However, a previous report in Ghana has shown that G6PD deficient males (A-) and females (A-A-) combined exhibited about 44% normal G6PD enzymatic activities whiles heterozygous A- showed 55% [17]. Homozygous A-A- G6PDd females had the strongest protection against malaria compared to either hemizygote A- G6PDd males or heterozygotes BA- or AA- G6PDd females. Although, the current data cannot explain why homozygous A-A- G6PDd exhibited a stronger protective effect than hemizygote A- G6PDd males, it may be due to the levels of 'self-limited haemolysis'. A similar observation was reported among persons with the G6PDd Mahidol variant, where females *with the* heterozygous *Mahidol* G6PDd variant had a 30% reduction in P. vivax infection. Whereas females with the homozygous G6PDd Mahidol variant and males with the hemizygous G6PDd Mahidol variant exhibited a 61% and 40% reduction in P. vivax infection respectively [52].

The underlying mechanisms offered by the G6PDd A- variants against severe malaria is unknown [16, 20, 27]. The result of this study agrees with the previously reported decrease in parasitization rate between 2–80 times lower in heterozygous G6PDd A- alleles compared to normal erythrocytes [53]. However, it is well established that G6PDd A- exhibits a reduced enzymatic activity up to about 10% for the regulation of excessive stress generated through PPP by the erythrocytic stage Plasmodium parasites [54, 55]. There is a possibility of high oxidation of glutathione (GSH) to decrease malaria parasite survival in G6PDd A- erythrocytes, even though in vitro results have yielded conflicting results [55]. Several factors such as the levels of G6PD enzyme activity, the method used to determine the deficiency and the small number of samples employed in previous studies could have resulted in the conflicting reports on the protection offered by G6PDd against malaria [10, 56].

The prevalence of non-falciparum malaria observed in this study was 1.22% for P. malariae distributed in 8 regions and 0.72% of P. ovale identified in 3 regions. Interestingly, no study has reported the association between G6PDd A- variants and the P. malariae or P. ovale malaria. Possibly due to the lower prevalence of these species of malaria parasites [57, 58]. The hypnozoites of P. ovale and P. vivax are responsible for the relapse of malaria infection months after complete treatment [59, 60]. Primaquine (PQ) is the recommended antimalarial used for clearing these hypnozoites, however, PQ may cause haemolysis in G6PD deficient individuals [9, 61, 62]. Although some studies have reported the association between G6PDd A- variants and protection against P. vivax malaria [60–63]. This study observed that P. malariae and P. ovale are more likely to infect the normal G6PD B variant compared with the G6PDd A variant. No infection of P. malariae or P. ovale was identified in samples from females with G6PDd (homozygous A-A- and heterozygous AA- G6PDd variants). There was, however, a single P. ovale infection detected in one male with the hemizygous G6PDd A- variant. Though the prevalence of P. malariae and P. ovale were too low to attain statistical power for the associations between P. ovale or P. malariae and G6PDd A- variants, the complete absence of infection in A-A- and AA- variants are suggestive of possible protection against the severe forms of this non-falciparum malaria in Ghana.

## Limitations

Malaria infection was determined using microscopy, as such infections containing submicroscopic densities of infection were not captured under the malaria infected group.

## Conclusions

This study has shown the nationwide distribution and prevalence of G6PD genotypic variants as well as the pattern of P. falciparum infectivity among individuals harboring different G6PD variants. The results showed that homozygous deficient female A-A-, hemizygous male A-, heterozygous female BA- and AA- G6PDd variants protect against malaria infections across all the study regions. Again, non-falciparum malaria, P. ovale and P. malariae were more prevalent in G6PD B variants with no infections identified in individuals with the A-A- and AA- variants. Suggesting that G6PDd A- variants similarly offer protection against P. ovale and P. malariae malaria in Ghana.

## Supporting information

**S1 Fig. Age associated selection of malaria influence the selection of G6PD genotypes. A.** Restricted cubic spline curve showing the effects of age categories on malaria infections and their effects on the selection of G6PD genotypic variants. The spline curve was generated with 20 points across the age categories ranging from 1 to 5 for >20 years, 15–19 years, 10–14 years,

5–9 years and 0–4 years respectively. The spline curve shows that the relationships among the age categories, malaria infection and G6PD genotypic variants non-linear association. **B**. Box and Whiskers plot showing minimum and maximum interpolation of an estimated least square fit for the age categories 5–9 years and 0–4 years between malaria prevalence and G6PD genotypic variants associations. **C.** Residual of malaria infections calculated across the categories with the corresponding G6PD genotypic variants. The residual for age category 0–4 years ranges from -2.07 to 0.26 from genotype B to genotype AA- of the G6PD genotypic variants to adjust the confounding factors associated with the age categories.
(DOCX)

**S1 File. List of Health facilities and institutions.** The list of health facilities used for the sample collection and data analysis are shown below.
(XLSX)

**S2 File Sections of the data analysed.**
(XLSX)

**S1 Table. Sensitivity and specificity of detecting malaria infection among G6PD genotypic variants using area under the curve of receiver-operation curves.**
(DOCX)

**S2 Table. ANOVA statistics comparing malaria positive, malaria negative and untested samples among the G6PD variants.**
(DOCX)

## Acknowledgments

The authors are grateful to all the study participants for willingly donating samples to be used for this study.

## Author Contributions

**Conceptualization:** Linda Eva Amoah, George Adu Asumah, Nana Yaw Peprah, Alexander Asamoah, Keziah Laurencia Malm.

**Formal analysis:** Linda Eva Amoah, Kwame Kumi Asare.

**Funding acquisition:** Keziah Laurencia Malm.

**Methodology:** Donu Dickson, Joana Abankwa, Abena Busayo, Dorcas Bredu, Sherifa Annan.

**Supervision:** Linda Eva Amoah.

**Validation:** Linda Eva Amoah.

**Writing – original draft:** Linda Eva Amoah, Kwame Kumi Asare.

**Writing – review & editing:** Linda Eva Amoah, Kwame Kumi Asare, Donu Dickson, Joana Abankwa, Abena Busayo, Dorcas Bredu, Sherifa Annan, George Adu Asumah, Nana Yaw Peprah, Alexander Asamoah, Keziah Laurencia Malm.

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
