## [Decision Letter · Decision Letter 0]

3 Aug 2021

PONE-D-21-20229

Genotypic glucose-6-phosphate dehydrogenase (G6PD) deficiency protects against Plasmodium falciparum infections in individuals living in Ghana

PLOS ONE

Dear Dr. Kumi,

Thank you for submitting your manuscript to PLOS ONE. After careful consideration, we feel that it has merit but does not fully meet PLOS ONE’s publication criteria as it currently stands. Therefore, we invite you to submit a revised version of the manuscript that addresses the points raised during the review process.

Kindly address all provided comments, specifically of reviewer one. kindly also seek statistical advice for the analysis of this article, some of the methods and analysis may require refinement. kindly have a native English speaker revise the language.

We look forward to receiving your revised manuscript.

Kind regards,

Benedikt Ley

Academic Editor

PLOS ONE

Journal Requirements:

2. Please include in your Methods section (or in Supplementary Information files) the participating hospitals/institutions.

3. You have indicated that informed consent was taken from the parents of all the children included in the study. Please could you provide some clarification on whether parental informed consent was taken from the the minors included in the study.

“No competing interest were made known.”

5. Please note that in order to use the direct billing option the corresponding author must be affiliated with the chosen institute. Please either amend your manuscript to change the affiliation or corresponding author, or email us at plosone@plos.org with a request to remove this option.

7. Your ethics statement should only appear in the Methods section of your manuscript. If your ethics statement is written in any section besides the Methods, please move it to the Methods section and delete it from any other section. Please ensure that your ethics statement is included in your manuscript, as the ethics statement entered into the online submission form will not be published alongside your manuscript.

8. We note that Figure 1 in your submission contain copyrighted images. All PLOS content is published under the Creative Commons Attribution License (CC BY 4.0), which means that the manuscript, images, and Supporting Information files will be freely available online, and any third party is permitted to access, download, copy, distribute, and use these materials in any way, even commercially, with proper attribution. For more information, see our copyright guidelines: http://journals.plos.org/plosone/s/licenses-and-copyright.

Reviewers' comments:

Reviewer's Responses to Questions

**Comments to the Author**

1. Is the manuscript technically sound, and do the data support the conclusions?

Reviewer #1: Partly

Reviewer #2: Partly

2. Has the statistical analysis been performed appropriately and rigorously? 

Reviewer #1: No

Reviewer #2: Yes

3. Have the authors made all data underlying the findings in their manuscript fully available?

Reviewer #1: Yes

Reviewer #2: Yes

4. Is the manuscript presented in an intelligible fashion and written in standard English?

Reviewer #1: Yes

Reviewer #2: Yes

5. Review Comments to the Author

Reviewer #1: Dear Authors,

I find your finding to be really interesting and important if done and analyzed properly. However, I found a lot of redundancy in the data presented and also few very confusing data/figures/tables.

Please clarify the following:

1. How do you enrolled randomly your samples from the main study of 19869 participants?

2. G6PD B genotype is actually wild type, as the authors have mentioned that this genotype has ‘complete’ G6PD activity – meaning, normal G6PD activity. So when this is considered as part of the G6PD genotype categorization, therefore, one must have in mind that this is the normal genotype. The G6PD A variant although a variant but has normal G6PD activity as well. Thus, it safe to say, BB, BA, AA, are normal. G6PD A- is considered as the variant that has less G6PD activity than A or B variant and thus can hemolyze upon oxidative trigger. Thus, please work around this facts when discussing about your results. so when we are discussing G6PD A- variant, then we are talking about G6PD deficiency.

3. No need to say percentage prevalence as normally prevalence is already in % as it is the proportion of the population having malaria or having G6PD deficiency.

4. Table 1 there is no data on mean of Age (SEM). To be more informative, the malaria data should be divided between male and female as perhaps this can explain why A-A- is more resistant to malaria infection compared to A-.

5. Figure 2A, the y-axis is not prevalence but number of subjects, while your x-axis should be malaria+ NOT microscopy +

6. Figure 2B, what does mean prevalence mean in this instance? I actually found the figure to be very confusing. Malaria+ within G6PD deficient subject OR malaria+ in 6108 total participants of the study? Again, y-axis should be number of participants.

7. Figure 2C, this is better to describe all those participants who have certain G6PD genotypes and malaria+ as well (Delete Figure 2A and B as these are redundant). Also please make stacked bars instead of the present bargraph, this way is easier to get your point across.

8. Figure 3D, y-axis can be Sensitivity (no need to add 100%) and y-axis as 1-Specificity.

9. Table 2A, why is there a - in front of the %? I think this number represents the percentage and if so, please be consistent with the rest of the table. Also please divide this by gender for easier reading: males (B,A or A-) and females (BB, BA, BA-, AA, AA-, and A-A-).

10. Table 2B: the number in bracket is the percentage?

11. Lines 203-206: “The odds of detecting microscopic malaria among the individuals' G6PDd genotypic variants showed B (p<0.0006), BB (p<0.0001) and BA (p<0.0001) genotypic variants influence malaria infections while A-A- genotypic variant protect individuals from malaria infection with OR (95%CI) of 0.0784 (0.0265-0.2319), p<0.0001 (Table 3).” The genotypes B, BB and BA do not influence malaria infection but rather do not have any protection against malaria infection, in comparison to A-A- that showed some degree of protection. Please change the wording here. However, how do you explain the male hemizygous A- (same low G6PD activity as female homozygous A-A-) that has higher OR than A-A- in detecting malaria in this variant? From this Table, it seems that hemizygous G6PD deficient male (A-) does not offer protection as it has the same OR as A, B, AA, and even has higher OR than BB, BA, and BA-?

12. Same question as before, in Line 212, "The mean (SEM) prevalence.." what does it mean?

13. Figure 3A - delete since this information is shown already in Fig 2C and Table 2B. Delete Fig 3B.

14. Line 568 “D) the plot of the adjusted p value…” please change to C) however, this figure should be deleted as again very redundant information and confusing to see.

15. Lines 219-222 “the mean difference (95% CI) for BA vs B (-546.7(-997.3-116.1), p=0.0064), B vs A-A- (527.3 (96.75-957.9), p=0.009), B vs AA- (468.3 (37.75-898.9), p=0.026; for males were B vs A- (435.7 (5.081-807.9), p=0.046 and BB vs A-A- (395.3 (-35.25-825.9), p=0.0896.” This observation cannot be made as there is no logic or order as to why you compare B to AA- or B to A-A- as this is normal male to heterozygous and homozygous G6PD deficient females. Also line 221, it is for females not males in comparing BB to A-A-.

16. Figure 4A - delete as this is shown already in Figure 4C.

17. Figure 4B is confusing, what does mean 1 and 2 mean?

18. Figure 5A, please use 3D histogram rather than you current graph for better grasp of which region has highest G6PD genotype of which kind.

19. Figure 5B, delete as this is shown in Table 2A already.

20. Supp Fig 1 A is pretty difficult to distinguish, perhaps different colour scheme? As for your x-axis, it's better to put actual age range rather than age category unless you state this in the legend.

21. Supp. Fig 1C, please explain why there are 2 sets of genotypes, are tehse based on different age categories? If so, please indicate in your graph.

22. Please explain why only homozygous deficient females have highest protection to malaria infection compared to hemizygous deficient males?

Reviewer #2: The manuscript reported a cross-sectional survey of G6PDd genotypic variants in Ghana. The objective of this study was to identify the prevalence and distribution of G6PD genotypic variants in Ghana as well as determine whether G6PDd genotypic variants protect against symptomatic malaria infection. A total of 6108 subjects were enrolled in the. The authors concluded that G6PDd genotypic variants, A-A-, AA- and A- protect against P. falciparum, P. ovale and P. malariae infection in Ghana.

Comment:

1. It will be useful to have some ideas about the G6PD ranges across the region? Is there any difference in the G6PD activities according to tribes?

2. While there is selective protection of G6PDd against severe malaria that may also extend to asymptomatic infection. A molecular test would be a useful addition as it offers better sensitivity over microscopy. The authors could have done a diagnostic PCR for Plasmodium species as they had DNA samples from DBS.

3. Do the authors have serological evidence to support their claim? Severely deficient or heterozygous females may not be sero naïve.

4. Studies conducted in other parts of the world could not document the protective effect of G6PD deficiency. I would suggest the authors discuss this aspect (e.g.

1. https://doi.org/10.1371/journal.pmed.1003576 ) in their discussion section.

5. Although provided separately, a title is missing for Table1, Table 3, and all figures.

6. I think the sample size is not sufficient to draw a conclusion that G6PDd genotypic variants, A-A-, AA- and A- protect against P. ovale and P. malariae infection in Ghana.

7. In some places scientific names were not italicized.

6. PLOS authors have the option to publish the peer review history of their article (what does this mean?). If published, this will include your full peer review and any attached files.

Reviewer #1: No

Reviewer #2: No

---

## [Author Response · Author response to Decision Letter 0]

20 Aug 2021

Reviewer #1: Dear Authors,

I find your finding to be really interesting and important if done and analyzed properly. However, I found a lot of redundancy in the data presented and also few very confusing data/figures/tables.

Please clarify the following:

1. How do you enrolled randomly your samples from the main study of 19869 participants?

Response: Stratified random sampling method was applied to the recruited samples where each study region was considered as an individual group and randomly selected from each region to ensure equal representation in the sampling for the study. A total sample of 6108 participants were randomly selected from all the regions and for use in this study.

2. G6PD B genotype is actually wild type, as the authors have mentioned that this genotype has ‘complete’ G6PD activity – meaning, normal G6PD activity. So when this is considered as part of the G6PD genotype categorization, therefore, one must have in mind that this is the normal genotype. The G6PD A variant although a variant but has normal G6PD activity as well. Thus, it safe to say, BB, BA, AA, are normal. G6PD A- is considered as the variant that has less G6PD activity than A or B variant and thus can hemolyze upon oxidative trigger. Thus, please work around this facts when discussing about your results. so when we are discussing G6PD A- variant, then we are talking about G6PD deficiency.

Response: We have revised the discussion based on the reviewer’s suggestions.

3. No need to say percentage prevalence as normally prevalence is already in % as it is the proportion of the population having malaria or having G6PD deficiency.

Response: All percentage prevalence has been changed to prevalence

4. Table 1 there is no data on mean of Age (SEM). To be more informative, the malaria data should be divided between male and female as perhaps this can explain why A-A- is more resistant to malaria infection compared to A-.

Response: We have included the Age (SEM) and the data has been divided into male and female.

5. Figure 2A, the y-axis is not prevalence but number of subjects, while your x-axis should be malaria+ NOT microscopy +

Response: Figure 2A has been deleted from the data.

6. Figure 2B, what does mean prevalence mean in this instance? I actually found the figure to be very confusing. Malaria+ within G6PD deficient subject OR malaria+ in 6108 total participants of the study? Again, y-axis should be number of participants.

Response: Figure 2B has been deleted

7. Figure 2C, this is better to describe all those participants who have certain G6PD genotypes and malaria+ as well (Delete Figure 2A and B as these are redundant). Also please make stacked bars instead of the present bargraph, this way is easier to get your point across.

Response: Figure 2 A and B has been deleted and the figure 2 C has been converted to stacked bars as suggested by the reviewer. Figure 2 C is now labelled as Figure 2 A.

8. Figure 3D, y-axis can be Sensitivity (no need to add 100%) and y-axis as 1-Specificity.

Response: The Figure 2D has been labelled as Figure 2 B and the y-axis is sensitivity and x-axis is 1-specificity

9. Table 2A, why is there a - in front of the %? I think this number represents the percentage and if so, please be consistent with the rest of the table. Also please divide this by gender for easier reading: males (B,A or A-) and females (BB, BA, BA-, AA, AA-, and A-A-).

Response: The table 2 A has be reorganized into male and females and all the percentages are now in brackets

10. Table 2B: the number in bracket is the percentage?

Response: Yes, the numbers in the brackets are percentages. We have revised the table to show the percentages.

11. Lines 203-206: “The odds of detecting microscopic malaria among the individuals' G6PDd genotypic variants showed B (p<0.0006), BB (p<0.0001) and BA (p<0.0001) genotypic variants influence malaria infections while A-A- genotypic variant protect individuals from malaria infection with OR (95%CI) of 0.0784 (0.0265-0.2319), p<0.0001 (Table 3).” The genotypes B, BB and BA do not influence malaria infection but rather do not have any protection against malaria infection, in comparison to A-A- that showed some degree of protection. Please change the wording here. However, how do you explain the male hemizygous A- (same low G6PD activity as female homozygous A-A-) that has higher OR than A-A- in detecting malaria in this variant? From this Table, it seems that hemizygous G6PD deficient male (A-) does not offer protection as it has the same OR as A, B, AA, and even has higher OR than BB, BA, and BA-?

Response: These sentences have been revised to “The homozygote G6PDd A-A- female exhibited stronger protection against malaria infections (OR (95% CI) =0.00784 (0.0265-0.2319), p<0.0001 compared to heterozygote G6PDd AA- female (OR (95% CI) =0.6741 (0.4878-0.9316),p=0.0169 and hemizygote G6PDd A- male (OR (95% CI) =0.764 (0.5818-1.0035), p=0.0453 (Table 3).”

12. Same question as before, in Line 212, "The mean (SEM) prevalence.." what does it mean? 

Response: These sentences have been deleted from the result

13. Figure 3A - delete since this information is shown already in Fig 2C and Table 2B. Delete Fig 3B.

Response: Figure 3A and 3B have been deleted

14. Line 568 “D) the plot of the adjusted p value…” please change to C) however, this figure should be deleted as again very redundant information and confusing to see. 

Response: The figure 3 C has been deleted. The whole figure three has been deleted 

15. Lines 219-222 “the mean difference (95% CI) for BA vs B (-546.7(-997.3-116.1), p=0.0064), B vs A-A- (527.3 (96.75-957.9), p=0.009), B vs AA- (468.3 (37.75-898.9), p=0.026; for males were B vs A- (435.7 (5.081-807.9), p=0.046 and BB vs A-A- (395.3 (-35.25-825.9), p=0.0896.” This observation cannot be made as there is no logic or order as to why you compare B to AA- or B to A-A- as this is normal male to heterozygous and homozygous G6PD deficient females. Also line 221, it is for females not males in comparing BB to A-A-.

Response: The comparison between male and female G6PD deficiency has been deleted as it has been suggested heterozygous and homozygous G6PD deficient females can not be directly be compared to hemizygote male

16. Figure 4A - delete as this is shown already in Figure 4C.

Response: Figure 4 A has been deleted 

17. Figure 4B is confusing, what does mean 1 and 2 mean?

Response: Figure 4 B has also been deleted 

18. Figure 5A, please use 3D histogram rather than you current graph for better grasp of which region has highest G6PD genotype of which kind.

Response: Figure 5 A has been replaced in 3D histogram

19. Figure 5B, delete as this is shown in Table 2A already.

Response: Figure 5 B has been deleted

20. Supp Fig 1 A is pretty difficult to distinguish, perhaps different colour scheme? As for your x-axis, it's better to put actual age range rather than age category unless you state this in the legend.

Response: We have stated the actual age ranges in the legends

21. Supp. Fig 1C, please explain why there are 2 sets of genotypes, are tehse based on different age categories? If so, please indicate in your graph.

Response: The second set of genotypes were interpolation of genotypes across the age groups. The plots of interpolation have been deleted from the graph.

22. Please explain why only homozygous deficient females have highest protection to malaria infection compared to hemizygous deficient males?

Response: Homozygous A-A- G6PDd females had the strongest protection against malaria compared to either hemizygote A- G6PDd males or heterozygotes BA- or AA- G6PDd females. Although, the current data cannot explain why homozygous A-A- G6PDd exhibited a stronger protective effect than hemizygote A- G6PDd males, it may be due to the levels of ‘self-limited haemolysis’. A similar observation was reported among persons with the G6PDd Mahidol variant, where females with the heterozygous Mahidol G6PDd variant had a 30% reduction in P. vivax infection. Whereas females with the homozygous G6PDd Mahidol variant and males with the hemizygous G6PDd Mahidol variant exhibited a 61% and 40% reduction in P. vivax infection respectively [52].

Reviewer #2: The manuscript reported a cross-sectional survey of G6PDd genotypic variants in Ghana. The objective of this study was to identify the prevalence and distribution of G6PD genotypic variants in Ghana as well as determine whether G6PDd genotypic variants protect against symptomatic malaria infection. A total of 6108 subjects were enrolled in the. The authors concluded that G6PDd genotypic variants, A-A-, AA- and A- protect against P. falciparum, P. ovale and P. malariae infection in Ghana.

Comment:

1. It will be useful to have some ideas about the G6PD ranges across the region? Is there any difference in the G6PD activities according to tribes?

Response: The global G6PDd prevalence based on DNA analysis is estimated to be around 7.1% with Africa estimated to have prevalence of 24%, American 2.9%, Asia 3.2%, Europe 4.8%, Middle East 3.8% and 0% in the Pacific [36]. In Ghana, it is estimated to be around 12.4% [17]. However, no report has been made on the individual tribes in Ghana. The result of this study showed that G6PDd genotypic variants were homogenous throughout Ghana. Thus, a similar distribution of G6PDd among the various tribes in Ghana.

2. While there is selective protection of G6PDd against severe malaria that may also extend to asymptomatic infection. A molecular test would be a useful addition as it offers better sensitivity over microscopy. The authors could have done a diagnostic PCR for Plasmodium species as they had DNA samples from DBS.

Response: unfortunately, we do not have such data for the current manuscript. We have inserted limitation section to this effect.

Limitations

Malaria infection was determined using microscopy, as such infections containing submicroscopic densities of infection were not captured under the malaria infected group.

3. Do the authors have serological evidence to support their claim? Severely deficient or heterozygous females may not be sero naïve.

Response: In this study, we only performed PCR genotyping for G6PD deficiency and do not have any phenotypic data. Although we have previously found evidence to support the fact that in some exceptional cases individuals with G6PD deficient genotypes do not have deficient/low enzyme activities

4. Studies conducted in other parts of the world could not document the protective effect of G6PD deficiency. I would suggest the authors discuss this aspect (e.g.

1. https://doi.org/10.1371/journal.pmed.1003576 ) in their discussion section.

Response: we have rewritten this section to include ‘Although there were significant associations between both the BA- and AA- heterozygous alleles and the protection against malaria [49], the AA- heterozygous allele offered stronger protection against malaria than the BA- allele. Previous reports that treated and analyzed the two female heterozygous G6PDd genotypes (BA- and AA-) as a combined genotype did not find any association between the female heterozygous G6PDd A- and protection against malaria [19, 20, 48].’

 5. Although provided separately, a title is missing for Table1, Table 3, and all figures.

Response: We have included the table titles and figure legends

 6. I think the sample size is not sufficient to draw a conclusion that G6PDd genotypic variants, A-A-, AA- and A- protect against P. ovale and P. malariae infection in Ghana.

Response: Although we did have a large sample size, due to the low prevalence of Po and Pm, we accept that the sample size of Po and Pm was low. 

We have revised the statement to ‘Suggesting that G6PDd A- variants may offer protection against P. ovale and P. malariae malaria in Ghana.’

7. In some places scientific names were not italicized.

Response: We have italicized all the scientific names

---

## [Decision Letter · Decision Letter 1]

6 Sep 2021

Genotypic glucose-6-phosphate dehydrogenase (G6PD) deficiency protects against Plasmodium falciparum infection in individuals living in Ghana

PONE-D-21-20229R1

Dear Dr. Kumi,

We’re pleased to inform you that your manuscript has been judged scientifically suitable for publication and will be formally accepted for publication once it meets all outstanding technical requirements.

Kind regards,

Benedikt Ley

Academic Editor

PLOS ONE

Additional Editor Comments (optional):

Reviewers' comments:

Reviewer's Responses to Questions

**Comments to the Author**

1. If the authors have adequately addressed your comments raised in a previous round of review and you feel that this manuscript is now acceptable for publication, you may indicate that here to bypass the “Comments to the Author” section, enter your conflict of interest statement in the “Confidential to Editor” section, and submit your "Accept" recommendation.

Reviewer #1: All comments have been addressed

Reviewer #2: All comments have been addressed

2. Is the manuscript technically sound, and do the data support the conclusions?

Reviewer #1: Yes

Reviewer #2: Yes

3. Has the statistical analysis been performed appropriately and rigorously? 

Reviewer #1: Yes

Reviewer #2: Yes

4. Have the authors made all data underlying the findings in their manuscript fully available?

Reviewer #1: Yes

Reviewer #2: Yes

5. Is the manuscript presented in an intelligible fashion and written in standard English?

Reviewer #1: Yes

Reviewer #2: Yes

6. Review Comments to the Author

Reviewer #1: Dear Authors,

I am pleased to see that the revised manuscript showed very good improvement in terms of clarity and statistical analysis.

I have spotted only 1 very minor mistake:

Line 116, delete "then" in ...the then 10 regions of Ghana...

I also would like to comment whether the reason you've shown very good protection against homozygous deficient females with A-A- genotype was because the number of subjects with this genotype is very small almost ten fold less than those with hemizygous deficient males A-?

Also, whether there are more chances of males being outside than females that you see this protection?

As I said before, your study is very interesting. It would be nice if you can pair this genotype with phenotype results as well.

Thank you.

Reviewer #2: (No Response)

7. PLOS authors have the option to publish the peer review history of their article (what does this mean?). If published, this will include your full peer review and any attached files.

Reviewer #1: No

Reviewer #2: No

---

## [Editor Report · Acceptance letter]

10 Sep 2021

PONE-D-21-20229R1 

Genotypic glucose-6-phosphate dehydrogenase (G6PD) deficiency protects against *Plasmodium falciparum* infection in individuals living in Ghana 

Dear Dr. Asare:

I'm pleased to inform you that your manuscript has been deemed suitable for publication in PLOS ONE. Congratulations! Your manuscript is now with our production department. 

Kind regards, 

on behalf of

Dr. Benedikt Ley 

Academic Editor

PLOS ONE